# PRETRAINED ENCYCLOPEDIA: WEAKLY SUPERVISED KNOWLEDGE-PRETRAINED LANGUAGE MODEL

**Wenhan Xiong**[†]**, Jingfei Du**[§]**, William Yang Wang**[†]**, Veselin Stoyanov**[§]**,**
[†] University of California, Santa Barbara
[§] Facebook AI
{xwhan, william}@cs.ucsb.edu, {jingfeidu, ves}@fb.com

## ABSTRACT

Recent breakthroughs of pretrained language models have shown the effectiveness of self-supervised learning for a wide range of natural language processing (NLP) tasks. In addition to standard syntactic and semantic NLP tasks, pretrained models achieve strong improvements on tasks that involve real-world knowledge, suggesting that large-scale language modeling could be an implicit method to capture knowledge. In this work, we further investigate the extent to which pretrained models such as BERT capture knowledge using a zero-shot fact completion task. Moreover, we propose a simple yet effective weakly supervised pretraining objective, which explicitly forces the model to incorporate knowledge about real-world entities. Models trained with our new objective yield significant improvements on the fact completion task. When applied to downstream tasks, our model consistently outperforms BERT on four entity-related question answering datasets (*i.e.*, WebQuestions, TriviaQA, SearchQA and Quasar-T) with an average 2.7 F1 improvements and a standard fine-grained entity typing dataset (*i.e.*, FIGER) with 5.7 accuracy gains.

## 1 INTRODUCTION

Language models pretrained on a large amount of text such as ELMo (Peters et al., 2018a)), BERT (Devlin et al., 2019) and XLNet (Yang et al., 2019c) have established new state of the art on a wide variety of NLP tasks. Researchers ascertain that pretraining allows models to learn syntactic and semantic information of language that is then transferred on other tasks (Peters et al., 2018b; Clark et al., 2019). Interestingly, pretrained models also perform well on tasks that require grounding language and reasoning about the real world. For instance, the new state-of-the-art for WNLI (Wang et al., 2019a), ReCoRD (Zhang et al., 2018) and SWAG (Zellers et al., 2018) is achieved by pretrained models. These tasks are carefully designed so that the text input alone does not convey the complete information for accurate predictions – external knowledge is required to fill the gap. These results suggest that large-scale pretrained models implicitly capture real-world knowledge. Logan et al. (2019) and Petroni et al. (2019) further validate this hypothesis through a zero-shot fact completion task that involves single-token entities, showing that pretrained models achieve much better performance than random guessing and can be on par with specifically-trained relation extraction models.

As unstructured text encodes a great deal of information about the world, large-scale pretraining over text data holds the promise of simultaneously learning syntax, semantics and connecting them with knowledge about the real world within a single model. However, existing pretraining objectives are usually defined at the token level and do not explicitly model entity-centric knowledge. In this work, we investigate whether we can further enforce pretrained models to focus on encyclopedic knowledge about real-world entities, so that they can better capture entity information from natural language and be applied to improving entity-related NLP tasks. We evaluate the extent to which a pretrained model represents such knowledge by extending an existing fact completion evaluation to a cloze ranking setting that allows us to deal with a large number of multi-token entity names without manual judgments. Our experiments on 10 common Wikidata (Vrandečić & Krötzsch, 2014) relations reveal that existing pretrained models encode entity-level knowledge only to a limited degree. Thus, we propose a new weakly supervised knowledge learning objective that requires the

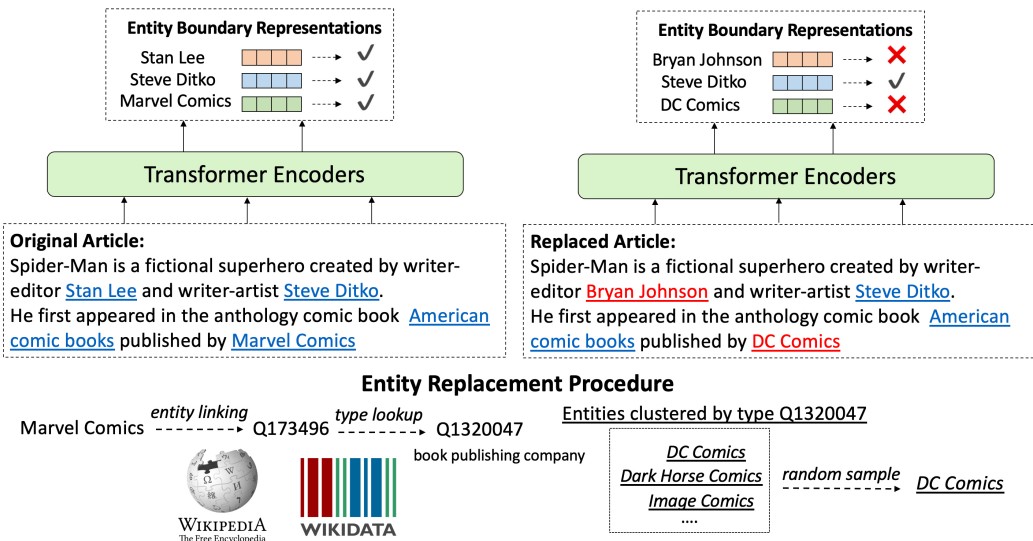

Figure 1: Type-Constrained Entity Replacements for Knowledge Learning.

model to distinguish between true and false knowledge expressed in natural language. Specifically, we replace entity mentions in the original documents with names of other entities of the same type and train the models to distinguish the correct entity mention from randomly chosen ones. Models trained with this objective demonstrates much stronger fact completion performance for most relations we test on. Compared with previous work (Zhang et al., 2019; Peters et al., 2019) that utilizes an external knowledge base to incorporate entity knowledge, our method is able to directly derive real-world knowledge from unstructured text. Moreover, our method requires no additional data processing, memory or modifications to the BERT model when fine-tuning for downstream tasks.

We test our model on two practical NLP problems that require entity knowledge: Question Answering (QA) and fine-grained Entity Typing. We use four previously published datasets for open-domain QA and observe that questions in these datasets often concern entities. The Entity Typing task requires the model to recognize fine-grained types of specified entity mentions given short contexts. On three of the QA datasets, our pretrained model outperforms all previous methods that do not rely on memory-consuming inter-passage normalizations[1]. On the FIGER entity-typing dataset, our model sets a new state of the art. Through ablation analysis, we show that the new entity-centric training objective is instrumental for achieving state-of-the-art results.

In summary, this paper makes the following contributions: 1) We extend existing fact completion evaluation settings to test pretrained models' ability on encoding knowledge of common real-world entities; 2) We propose a new weakly supervised pretraining method which results in models that better capture knowledge about real-world entities from natural language text; 3) The model trained with our knowledge learning objective establishes new state of the art on three entity-related QA datasets and a standard fine-grained entity typing dataset.

We begin by introducing our weakly supervised method for knowledge learning (§2) and then discuss experiment settings and evaluation protocols, compare our model to previously published work and perform ablation analysis. Finally, we review related work in §4 and conclude in §5.

## 2 ENTITY REPLACEMENT TRAINING

We design an entity-centric training objective that utilizes weakly supervised training signals to explicitly encourage knowledge learning during pretraining. Given an input document, we first

---

[1]Wang et al. (2019b) propose to use BERT to encode multiple retrieved paragraphs within the same forward-backward pass. This operation requires around 12G GPU memory even for encoding a single example which includes 6 512-token paragraphs.

recognize the entity mentions and link them to Wikipedia entities[2]. We consider the original texts as positive knowledge statements and create negative statements by randomly replacing the entity mentions ($\mathcal{E}^+$) with the names of other random entities ($\mathcal{E}^-$) that have the same entity type as the mentioned entity. This setup is similar in spirit to the type-constrained negative sampling technique used to train knowledge base representations (Bordes et al., 2013). The latter technique creates negative triples by replacing the subject or object entity with random entities of the same type. Instead of knowledge base triples, we treat unstructured texts as factual statements. For a certain entity $e$ mentioned in a context $\mathcal{C}$, we train the model to make a binary prediction indicating whether the entity has been replaced:

$$J_{e,\mathcal{C}} = \mathbb{1}_{e \in \mathcal{E}^+} \log P(e|\mathcal{C}) + (1 - \mathbb{1}_{e \in \mathcal{E}^+}) \log(1 - P(e|\mathcal{C})).$$

Compared to the language modeling objective, entity replacement is defined at the entity level and introduces stronger negative signals. When we enforce entities to be of the same type, we preserve the linguistic correctness of the original sentence while the system needs to learn to perform judgment based on the factual aspect of the sentence.

We describe the implementation in more detail in the following paragraphs.

**Data Preparation**  We use the whole English Wikipedia dump as training data and rely on all Wikipedia entities[3]. Entities in documents are recognized based on Wikipedia anchor links and entity alias from Wikidata. That is, we first retrieve the entities annotated by anchor links and then find other mentions of these entities by string matching their Wikidata alias. We split each document into multiple text chunks with the same size (512 tokens). Although our experiments rely on the Wikipedia corpus, this setup can be easily extended to larger corpora with off-the-shelf entity linking tools. We leave the larger scope of the experiments to future work.

**Replacement Strategy**  When replacing entities, we first lookup type information[4] from Wikidata and then randomly select other entities with the same type. We do not replace adjacent entities. In other words, there must be at least one unreplaced entity between any two replaced ones. This reduces cases where we replace all entities in the same sentence and the resulting sentences happen to introduce correct entities by chance. For replacement, we randomly sample a string from the entities' alias set. For each text chunk, we replicate it 10 times with different negative entities for each replacement location. We show an illustration of the entity replacement method in Figure 1.

**Model Architecture**  We use the Transformer (Vaswani et al., 2017) model used by BERT (Devlin et al., 2019). We use the same architecture as BERT base: 12 Transformer layers, each with hidden dimension 768. We initialize the transformer with a model pretrained based on our own BERT re-implementations[5]. For each entity, we use the final representations of its boundary words (words before and after the entity mention) to make predictions. We simply concatenate the boundary words' representations and add a linear layer for prediction. During training, we use 0.05 dropout at the final layer.

**Training Objectives**  Masked language model pretraining has been proven to be effective for downstream tasks. While training for entity replacement we also train with the masked language model objective in a multi-task set-up. When masking tokens, we restrict the masks to be outside the entity spans. We use a masking ratio of $5\%$ instead of $15\%$ in the original BERT to avoid masking out too much of the context. We train the model for approximately 1 million updates using a batch size of 128.

---

[2]The entity links are only required for pretraining.

[3]Each Wikipedia entity (title) corresponds to a unique entity node in Wikidata.

[4]Based on the "instance of" relation in Wikidata. If there are multiple correct types, we randomly choose one.

[5]We use the masked language model implementation in Fairseq (Ott et al., 2019) to pre-train model for 2M updates on the combination of BooksCorpus (Zhu et al., 2015) and English Wikipedia.

## 3 EXPERIMENTS

We first test our model on a fact completion task. This task resembles traditional knowledge base completion: it requires the model to complete missing entities in factual triples. We further test on two real-world downstream tasks that require entity-level knowledge – question answering and fine-grained entity typing. We describe the hyperparameter and training settings of all experiments in the appendix.

### 3.1 ZERO-SHOT FACT COMPLETION

In traditional knowledge base completion tasks models have access to a set of training triples. Instead, we utilize a zero-shot test to examine the model's ability to automatically derive relational knowledge from natural language.

**Dataset**   We rely on factual triples from Wikidata. Each triple describes the relationship between two certain entities, *e.g.*, {*Paris, CapitalOf, France*}. Following recent practices (Bosselut et al., 2019; Logan et al., 2019) that decode structured knowledge from language models, we first manually create templates to convert triples of 10 common relations into natural language expressions ({*Paris, CapitalOf, France*} → *the capital of France is Paris*). We then create queries by removing the object entity in the expression and use pre-trained models to predict the missing entities, *e.g.*, *the capital of France is ?*. We create 1000 cloze examples[6] for each of the 10 relations.

**Evaluation Metrics**   Previous work (Logan et al., 2019; Petroni et al., 2019) either relies on human evaluation or only considers single-token entities for fact completion. In contrast, we consider an entity-ranking setup and create a set of candidate entities for each relation. This setting allows us to automatically evaluate a large number of queries that usually involve multi-token entities. We test pretrained models on their ability to recover the correct object entity from the candidate set. To create the negative choices, we select from the set of all object entities in the particular relation, which generally have the same type as the groundtruth and are more challenging to distinguish than entities with different types. Our evaluation strategy is similar to previous work on knowledge base completion (Nickel et al., 2011; Bordes et al., 2013; Xiong et al., 2017). We follow these studies and use Hits@10 as the evaluation metric.

**Baselines**   We compare our model with two pretrained language models BERT (Devlin et al., 2019) (both base and large) and GPT-2 (Radford et al., 2019). We make use of their output token probabilities to rank candidate entities. For BERT, we feed in the masked queries (*e.g.*, $Q_{masked}$ = `the capital of France is [MASK]`). For multi-token candidates, we use the same number of `[MASK]` tokens in the query inputs. We use the average log probability of masked tokens for ranking. Given a multi-token entity $E_i = [e_i^1, e_i^2, ..., e_i^{|E_i|}]$, the ranking score from BERT is calculated as

$$S_{E_i} = \frac{1}{|E_i|} \sum_k \log P(e_i^k | Q_{masked}).$$

For GPT-2, we feed in the original query without the answer entity and use the first-token probability of candidate entities for ranking, which performs better than using average log probabilities. As our model learns to predict a plausible probability ($P(e|\mathcal{C})$) for each entity mention during entity replacement training, we can directly use these predicted probabilities to rank the candidates.

**Results**   Table 1 shows the fact completion results for all relations. We denote our method **WKLM** for (**W**eakly Supervised **K**nowledge-Pretrained **L**anguage **M**odel). Overall, WKLM achieves the best results on 8 of the 10 relations. We also observe that GPT-2 outperforms BERT on average. We think this is because the fact completion task requires models to predict the missing entities using only a short context on the left, while BERT pretraining incorporates context from both directions. Interestingly, BERT achieves good performance on several geographical relations such as `PlaceOfBirth`, `LocatedIn` and `PlaceOfDeath`. We conjecture that this is because location entities usually appear at sentence ends in Wikipedia articles, *e.g.*, `Obama was born in`

---

[6]For each relation, we use the top triples that connecting most common entities.

`Honolulu, Hawaii..` This sentence pattern is similar to our templates and BERT may learn to rely mostly on the left context to make predictions. For most relations that include answers that are person names, BERT lags behind both GPT-2 and our model.

Comparing the top and bottom five relations, we observe that BERT's performance is correlated with the size of the candidate set, while WKLM and GPT-2 are less sensitive to this number. A similar pattern exists between models' performance and the cardinality of groundtruth answers, *i.e.*, our model achieves similar performance on both single-answer and multiple-answer queries while BERT is usually better at single-answer queries. WKLM both outperforms BERT and GPT-2 and achieves robust performance across relations with different properties. Visualization of correlations between relation properties and model performance can be found in the appendix.

Table 1: Zero-Shot Fact Completion Results.

| Relation Name | # of Candidates | # of Answers | Model | | | |
|---|---|---|---|---|---|---|
| | | | **BERT-base** | **BERT-large** | **GPT-2** | **Ours** |
| HASCHILD (P40) | 906 | 3.8 | 9.00 | 6.00 | 20.5 | **63.5** |
| NOTABLEWORK (P800) | 901 | 5.2 | 1.88 | 2.56 | 2.39 | **4.10** |
| CAPITALOF (P36) | 820 | 2.2 | 1.87 | 1.55 | 15.8 | **49.1** |
| FOUNDEDBY (P112) | 798 | 3.7 | 2.44 | 1.93 | 8.65 | **24.2** |
| CREATOR (P170) | 536 | 3.6 | 4.57 | 4.57 | 7.27 | **9.84** |
| PLACEOFBIRTH (P19) | 497 | 1.8 | 19.2 | **30.9** | 8.95 | 23.2 |
| LOCATEDIN (P131)) | 382 | 1.9 | 13.2 | 52.5 | 21.0 | **61.1** |
| EDUCATEDAT (P69) | 374 | 4.1 | 9.10 | 7.93 | 11.0 | **16.9** |
| PLACEOFDEATH (P20) | 313 | 1.7 | **43.0** | 42.6 | 8.83 | 26.5 |
| OCCUPATION (P106) | 190 | 1.4 | 8.58 | **10.7** | 9.17 | **10.7** |
| Average Hits@10 | - | - | 11.3 | 16.1 | 16.3 | **28.9** |

## 3.2 DOWNSTREAM TASKS

Background knowledge is important for language understanding. We expect our pretraining approach to be beneficial to NLP applications where entity-level knowledge is essential. We consider two such applications: question answering and entity-typing. We find that a large portion of the questions in existing QA datasets are about entities and involve entity relations. In a way, our pretraining objective is analogous to question answering in a multiple-choice setting (Hermann et al., 2015). The entity-typing task requires the model to predict a set of correct types of entity mentions in a short context. The context itself can be insufficient and the training data for this task is small and noisy. We believe a model that encodes background entity knowledge can help in both cases.

### 3.2.1 QUESTION ANSWERING

**Datasets** We consider four question answering datasets:

- **WebQuestions** (Berant et al., 2013) is originally a dataset for knowledge base question answering. The questions are collected using Google Suggest API and are all asking about simple relational facts of Freebase entities.
- **TriviaQA**[7] (Joshi et al., 2017) includes questions from trivia and quiz-league websites. Apart from a small portion of questions to which the answers are numbers and free texts, $92.85\%$ of the answers are Wikipedia entities.
- **Quasar-T** (Dhingra et al., 2017) is another dataset that includes trivia questions. Most of the answers in this dataset are none phrases. According to our manual analysis on random samples, $88\%$ of the answers are real-world entities[8].
- **SearchQA** (Dunn et al., 2017) uses questions from the television quiz show *Jeopardy!* and we also find that almost all of the answers are real-world entities.

---

[7]The splits of TriviaQA might be different in previous work, we use the same splits used by Lin et al. (2018). Different methods also use different retrieval systems for this dataset, *i.e.*, ORQA and BERTserini retrieve documents from Wikipedia while the other methods use retrieved Web documents.

[8]We consider answers as entities as long as they correspond to Wikidata entities or Wikipedia titles.

Questions in all three datasets are created without the context of a paragraph, which resembles the scenario of practical question answering applications. All the questions except WebQuestions are written by humans. This indicates that humans are generally interested to ask questions to seek information about entities. We show the statistics and example questions in Table 2. We split the training data (created by distant supervision) of WebQuestions with a ratio (9:1) for training and development. Since our model is based on our own BERT implementations, in addition to the aforementioned entity-related datasets, we first use the standard SQuAD (Rajpurkar et al., 2016) benchmark to validate our model's answer extraction performance.

Table 2: Properties of the QA Datasets.

| Dataset | Train | Valid | Test | Example Questions |
|---------|-------|-------|------|-------------------|
| WebQuestions | 3778 | - | 2032 | *Who plays Stewie Griffin on Family Guy?* |
| TriviaQA | 87291 | 11274 | 10790 | *What is the Japanese share index called?* |
| SearchQA | 99811 | 13893 | 27247 | *Hero several books 11 discover's wizard?* |
| Quasar-T | 37012 | 3000 | 3000 | *Which vegetable is a Welsh emblem?* |

**Settings** We adopt the fine-tuning approach to extract answer spans with pretrained models. We add linear layers over the last hidden states of the pretrained models to predict the start and end positions of the answer. Unlike SQuAD, questions in the datasets we use are not paired with paragraphs that contain the answer. We follow previous work (Chen et al., 2017; Wang et al., 2018a) and retrieve context paragraphs with information retrieval systems. Details of the context retrieval process for each dataset can be found in the appendix. Reader models are trained with distantly supervised data, *i.e.*, we treat any text span in any retrieved paragraph as ground truth as long as it matches the original answers. Since the reader model needs to read multiple paragraphs to predict a single answer at inference time, we also train a BERT based paragraph ranker with distant-supervised data to assign each paragraph a relevance score. The paragraph ranker takes question and paragraph pairs and predicts a score in the range $[0, 1]$ for each pair. During inference, for each question and its evidence paragraph set, we first use the paragraph reader to extract the best answer from each paragraph. These answers are then ranked based on a linear combination of the answer extraction score (a log sum of the answer start and end scores) and the paragraph relevance score. We also evaluate model performance without using the relevance scores.

**Open-Domain QA Baselines** We compare our QA model with the following systems:

- **DrQA** (Chen et al., 2017) is an open-domain QA system which uses TF-IDF with bigram features for ranking and a simple attentive reader for answer extraction.
- **R$^3$** (Wang et al., 2018a) is a reinforcement learning based system which jointly trains a paragraph ranker and a document reader.
- **DSQA** (Lin et al., 2018) uses RNN-based paragraph ranker and jointly trains the paragraph ranker and attentive paragraph ranker with a multi-task loss.
- **Evidence Aggregation** (Wang et al., 2018b) uses a hybrid answer reranking module to aggregate answer information from multiple paragraphs and rerank the answers extracted from multiple paragraphs.
- **BERTserini** (Yang et al., 2019a) is a BERT-based open-domain QA system, which uses BM25-based retriever to retrieve 100 paragraphs and a BERT-based reader to extract answers. The paragraph reader is either trained with SQuAD (Rajpurkar et al., 2016) data or distant-supervision data (Yang et al., 2019b)
- **ORQA** (Lee et al., 2019) replaces the traditional BM25 ranking with a BERT-based ranker. The ranker model is pretrained on the whole Wikipedia corpus with an inverse cloze task which simulates the matching between questions and paragraphs. All text blocks in Wikipedia are be pre-encoded as vectors and retrieved with Locality Sensitive Hashing.

**Results** Table 3 shows the SQuAD results and Table 4 shows the open-domain results on the four datasets that are highly entity-related. From the SQuAD results, we observe that our BERT reimplementation performs better than the original model this is due to the fact that it is trained

for twice as many updates: 2 million vs. 1 million for the original BERT. Although lots of the answers in SQuAD are non-entity spans, the WKLM model we propose achieves better performance than BERT. We believe the improvement is due to both the masked language model and entity replacement objectives. Ablation experiments on the training objectives will be discussed in §3.2.3.

Having established that our BERT re-implementation performs better than the original model, we compare with only our own BERT for the following experiments. From Table 4, we see that our model produces consistent improvements across different datasets. Compared to the 0.8 F1 improvements over BERT on SQuAD, we achieve an average of 2.7 F1 improvements over BERT on entity-related datasets when the ranking scores are not used. On TriviaQA and Quasar-T, WKLM outperforms our BERT even when it uses ranking scores. Improvements in natural language question datasets (WebQuestions, TriviaQA, and Quasar-T) are more significant than SearchQA where the questions are informal queries. When we utilize ranking scores from a simple BERT based ranker, we are able to achieve the state-of-the-art on three of the four datasets.

Table 3: SQuAD Dev Results.

| Model | EM | F1 |
|---|---|---|
| Google's BERT-base | 80.8 | 88.5 |
| Google's BERT-large | 84.1 | 90.9 |
| Our BERT-base | 83.4 | 90.5 |
| WKLM (base) | 84.3 | 91.3 |

Table 4: Open-domain QA Results.

| Model | WebQuestions | | TriviaQA | | Quasar-T | | SearchQA | |
|---|---|---|---|---|---|---|---|---|
| | EM | F1 | EM | F1 | EM | F1 | EM | F1 |
| DrQA (Chen et al., 2017) | 20.7 | - | - | - | - | - | - | - |
| R$^3$ (Wang et al., 2018a) | - | - | 50.6 | 57.3 | 42.3 | 49.6 | 57.0 | 63.2 |
| DSQA (Lin et al., 2018) | 18.5 | 25.6 | 48.7 | 56.3 | 42.2 | 49.3 | 49.0 | 55.3 |
| Evidence Agg. (Wang et al., 2018b) | - | - | 50.6 | 57.3 | 42.3 | 49.6 | 57.0 | 63.2 |
| BERTserini (Yang et al., 2019a) | - | - | 51.0 | 56.3 | - | - | - | - |
| BERTserini+DS (Yang et al., 2019b) | - | - | 54.4 | 60.2 | - | - | - | - |
| ORQA (Lee et al., 2019) | **36.4** | - | 45.0 | - | - | - | - | - |
| Our BERT | 29.2 | 35.5 | 48.7 | 53.2 | 40.4 | 46.1 | 57.1 | 61.9 |
| Our BERT + Ranking score | 32.2 | 38.9 | 52.1 | 56.5 | 43.2 | 49.2 | 60.6 | 65.9 |
| WKLM | 30.8 | 37.9 | 52.2 | 56.7 | 43.7 | 49.9 | 58.7 | 63.3 |
| WKLM + Ranking score | 34.6 | 41.8 | **58.1** | **63.1** | **45.8** | **52.2** | **61.7** | **66.7** |

### 3.2.2 ENTITY TYPING

To compare with an existing study (Zhang et al., 2019) that also attempts to incorporate entity knowledge into language models, we consider an additional entity typing task using the large FIGER dataset (Ling & Weld, 2012). The task is to assign a fine-grained type to entity mentions. We do that by adding two special tokens before and after the entity span to mark the entity position. We use the final representation of the start token ([CLS]) to predict the entity types. The model is fine-tuned on weakly-supervised training data with binary cross-entropy loss. We evaluate the models using strict accuracy, loose micro, and macro F1 scores.

We show the results in Table 5. We compare our model with two non-BERT neural baselines (Inui et al., 2017) that integrate a set of hand-crafted features: **LSTM + Hand-crafted** and **Attentive +**

Table 5: Fine-grained Entity Typing Results on the FIGER dataset.

| Model | Acc | Ma-F1 | Mi-F1 |
|---|---|---|---|
| LSTM + Hand-crafted (Inui et al., 2017) | 57.02 | 76.98 | 73.94 |
| Attentive + Hand-crafted (Inui et al., 2017) | 59.68 | 78.97 | 75.36 |
| BERT baseline (Zhang et al., 2019) | 52.04 | 75.16 | 71.63 |
| ERNIE (Zhang et al., 2019) | 57.19 | 75.61 | 73.39 |
| Our BERT | 54.53 | 79.57 | 74.74 |
| WKLM | **60.21** | **81.99** | **77.00** |

**Hand-crafted**; a vanilla **BERT baseline** and the **ERNIE** model (Zhang et al., 2019) that enhances BERT with knowledge base embeddings.

First, we see that naively applying BERT is less effective than simple models combined with sparse hand-crafted features. Although the ERNIE model can improve over BERT by 5.15 points, its performance still lags behind models that make good use of hand-crafted features. In contrast, although based on a stronger BERT model, our model achieves larger absolute improvements (5.68 points) and sets a new state-of-the-art for this task. Given the larger improvement margin, we believe our model that directly learn knowledge from text is more effective than the ERNIE method.

### 3.2.3 ABLATION STUDY: THE EFFECT OF MASKED LANGUAGE MODEL LOSS

In view of a recent study (Liu et al., 2019b) showing simply extending the training time of BERT leads to stronger performance on various downstream tasks, we conduct further analysis to differentiate the effects of entity replacement training and masked language modeling. We compare our model with three variants: a model pretrained only with the knowledge learning objective (**WKLM without MLM**), a model trained with both knowledge learning and masked language modeling with more masked words (**WKLM with** 15% **MLM**) and a BERT model trained with additional 1 million updates on English Wikipedia (**BERT + 1M MLM updates**) and no knowledge learning.

The ablation results are shown in Table 6. The results of WKLM without MLM validate that adding the language model objective is essential for downstream performance. We also find that masking out too many words (*i.e.*, 15% masking ratio as in the original BERT) leads to worse results. We conjecture that too many masked words outside entity mentions break parts of the context information and introduce noisy signals to knowledge learning. Results of continued BERT training show that more MLM updates are often beneficial, especially for SQuAD. However, on tasks that are more entity-centric, continued MLM training is less effective than our WKLM method. This suggests that our WKLM method could serve as an effective complementary recipe to masked language modeling when applied to entity-related NLP tasks.

Table 6: Ablation Studies on Masked Language Model and Masking Ratios.

| Model | SQuAD | | TriviaQA | | Quasar-T | | FIGER |
|---|---|---|---|---|---|---|---|
| | EM | F1 | EM | F1 | EM | F1 | Acc |
| Our BERT | 83.4 | 90.5 | 48.7 | 53.2 | 40.4 | 46.1 | 54.53 |
| WKLM | 84.3 | **91.3** | **52.2** | **56.7** | **43.7** | **49.9** | **60.21** |
| WKLM without MLM | 80.5 | 87.6 | 48.2 | 52.5 | 42.2 | 48.1 | 58.44 |
| WKLM with 15% masking | 84.1 | 91.0 | 51.0 | 55.3 | 42.9 | 49.0 | 59.68 |
| Our BERT + 1M MLM updates | **84.4** | 91.1 | 52.0 | 56.3 | 42.3 | 48.2 | 54.17 |

## 4 RELATED WORK

**Pretrained Language Representations**  Early research on language representations focused on static unsupervised word representations (Mikolov et al., 2013; Pennington et al., 2014). Word embeddings leverage co-occurrences to learn latent word vectors that approximately reflect word semantics. Given that words can have different meanings in different contexts, more recent studies (McCann et al., 2017; Peters et al., 2018a) show that contextual language representations can be more powerful than static word embeddings in downstream tasks. This direction has been further explored at a larger scale with efficient Transformer architectures (Radford et al., 2019; Devlin et al., 2019; Yang et al., 2019c). Our WKLM method is based on these techniques and we focus on improving the knowledge ability of pretrained models.

**Knowledge-Enhanced NLP Models**  Background knowledge has been considered an indispensable part of language understanding (Fillmore et al., 1976; Minsky, 1988). As standard language encoders usually do not explicitly model knowledge, recent studies (Ahn et al., 2016; Yang & Mitchell, 2017; Logan et al., 2019; Liu et al., 2019a) have explored methods to incorporate external knowledge into NLP models. Most of these methods rely on additional inputs such as entity representations from structured knowledge bases. With the breakthrough of large-scale pretrained language

encoders (Devlin et al., 2019), Zhang et al. (2019) and Peters et al. (2019) adopt similar ideas and propose entity-level knowledge enhancement training objectives to incorporate knowledge into pretrained models. Other recent studies (Mihaylov & Frank, 2018; Xiong et al., 2019) leverage external knowledge bases to enhance text-based question answering models. In contrast to these methods, our method utilizes minimal external entity information and does not require additional memory or architectural changes when applied to downstream tasks.

## 5 CONCLUSION

We introduce a weakly supervised method to encourage pretrained language models to learn entity-level knowledge. Our method uses minimal entity information during pretraining and does not introduce additional computation, memory or architectural overhead for downstream task fine-tuning. The trained model demonstrates strong performance on a probing fact completion task and two entity-related NLP tasks. Together, our results show the potential of directly learning entity-level knowledge from unstructured natural language and the benefits of large-scale knowledge-aware pretraining for downstream NLP tasks.

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

## A  APPENDIX

**Implementation Details and Hyperparameters**   We implement our method using Fairseq Ott et al. (2019) and the fact completion baselines are implemented with Huggingface's Pytorch-Transformers[9]. We pretrain the models with 32 V100 GPUs for 3 days. We use at most 2 GPUs for fine-tuning the paragraph reader, use 8 GPUs for fine-tuning the paragraph ranker. The entity-typing experiments require larger batch sizes and take 8 GPUs for training.

For the knowledge learning pretraining phase, we use the Adam optimizer (Kingma & Ba, 2014) with learning rate 1e-5, batch size 128 and weight decay 0.01. The model is pretrained on 32 V100 GPUs for 3 days. To train the paragraph reader for open-domain QA, we select the best learning rate from {1e-6, 5e-6, 1e-5, 2e-5} and last layer dropout ratio from {0.1, 0.2}. We set the maximum training epoch to be 10 and batch size to be 32. The maximal input sequence length is 512 for WebQuestions and 128 for the other three datasets that use sentence-level paragraphs. For the paragraph ranker, we choose learning rate from {1e-5, 2e-5, 5e-6}, use dropout 0.1 and batch size 256. The maximal sequence length for each dataset is consistent with the one we used for training the paragraph reader. The linear combination of ranking and extraction scores is selected based on validation performance. For SQuAD experiments, we select learning rate from {1e-5, 5e-6, 2e-5, 3e-5}, learning rate from {8, 16}, last layer dropout ratio from {0.1, 0.2}. We set the maximal sequence length as 512 and the maximal training epoch as 5. For entity typing, we select learning rate from {1e-5, 2e-5, 3e-5, 5e-5} and batch size from {128, 256}. We set the maximal sequence length to be 256, the last layer dropout ratio to be 0.1. The model is fine-tuned for at most 3 epochs to prevent overfitting. The threshold for type prediction is selected on the validation set.

**Context Collection for QA Datasets**   For WebQuestions, we collect evidence context using the document retriever of DrQA (Chen et al., 2017), which uses TF-IDF based metric to retrieve the top 5 Wikipedia articles. For Quasar-T, we use Lucene ranked paragraphs. For SearchQA and TriviaQA, we use paragraphs ranked by search engines. Following existing research (Wang et al., 2018b; Lin et al., 2018), we use sentence-level paragraphs for SearchQA (50 sentences), TriviaQA (100 sentences) and SearchQA (100 sentences).

**Correlation between Fact Completion Results and Properties of Relations**   Figure 2 shows the fact completion results of BERT are unstable on different relations with different properties, *i.e.*, BERT's performance is strongly correlated with the size of candidate entity set and the number of groundtruth answers. Compared to BERT, WKLM is often less sensitive to these two factors.

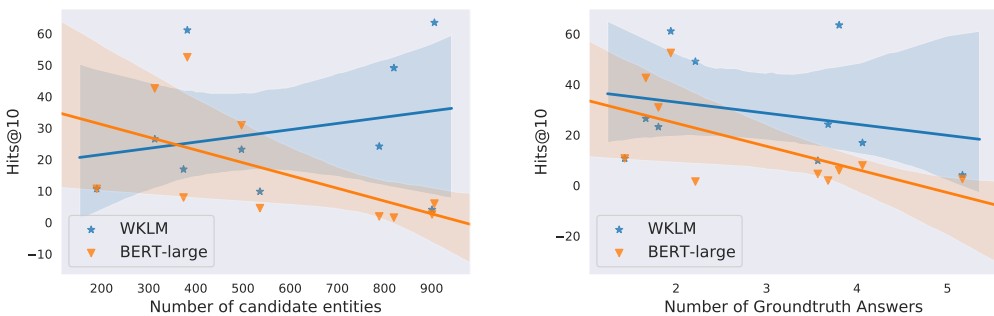

Figure 2: **Left:** Correlation between candidate set size and hits@10; **Right:** Correlation between number of groundtruth answers and hits@10.

---

