# OpenReview forum: "Pretrained Encyclopedia: Weakly Supervised Knowledge-Pretrained Language Model"
_ICLR.cc/2020/Conference — Accept (Poster)_

### Official Review · AnonReviewer2 · 2019-10-20
**Official Blind Review #2**

**Rating:** 8

**Review:**

This paper aims to incorporate world knowledge for the pretraining approach so that (1) pretrained models contain useful information about the world, and (2) benefit downstream NLP tasks. The paper does so by introducing the objective which distinguishes the groundtruth entity and the false entity in the Wikipedia text. This was carefully done by detecting entities in the text, find the corresponding entity in Wikidata, randomly choose another entity which has the same type as the original entity, and make sure this doesn’t happen for neighboring entities or too much in order to avoid context change. Adding this objective to the original masked LM objective, this pretrained model is shown to be effective and outperform baselines significantly in many tasks such as zero-shot fact completion, question answering, and fine-grained entity typing.

Particularly, I appreciate (1) the fact that they compare with other knowledge-aware pretrained models such as ERNIE, and (2) their ablation in Table 6 which compares only knowledge learning objective, more masking or finetuning with knowledge learning objective starting from MLM instead of multi-task learning (actually, is BERT+1M MLM updates a correct term? Since MLM is masked LM, shouldn’t it be BERT+1M knowledge learning?): it clearly shows that it is important to do both MLM and knowledge learning, appropriate ratio for masking, and multi-task learning between MLM and knowledge learning instead of some kind of pretrain-finetune approach.

Some marginal concerns I have is that some settings for downstream tasks are not clear. For example, for WebQuestions the number of validation examples is not specified, but I believe they should have split the train set into train/valid for development (such as early stopping or hyperparameter tuning), and conventionally people have split the train set into 90/10 for train/valid. In addition, as far as I know, TriviaQA has a bunch of settings such as Wiki setting, Web setting, unfiltered setting and open setting. What exactly is this setting? The paper mentions they follow Lin et al. 2018, but the statistics shown in Table 2 are different from their statistics. In addition, the authors compare results in different settings in Table 4 (for example, ORQA is in open setting, while I think the setting in this paper is not).

Some clarification questions:

1. I understand Wikipedia anchor links means hyperlinks. How do you know that the mention with a hyperlink is always an entity? Also, there are many cases that the mention with the hyperlink does not match in meaning with the linked article. For example, in “https://en.wikipedia.org/wiki/Barack_Obama”, there is a sentence “He was elected over Republican John McCain and was inaugurated on January 20, 2009” where “elected” is linked to “https://en.wikipedia.org/wiki/2008_United_States_presidential_election” which is probably not what we want.

2. If there are same mentions in the text chunk, are all of them replaced? Probably with the same negative entity?

3. How often were entities to replace chosen? E.g. is one entity in the text chunk replaced at a time? Or multiple ones?

4. How do you look up the type of the entity? For instance, in the example of “Marvel Comics” illustrated in Figure 1, I see there are two triples, <Marvel Comics, instance of, business> and <Marvel Comics, instance of, book publishing company>. Is "instance of" used? How did you choose “book publishing company” over “business”? Or is it randomly chosen? I think ideally more fine-grained type should be chosen, but wonder if there is a way to find which one is more fine-grained.

5. Perhaps not necessary, but I wonder what happens if an entity is detected, and there is another masked LM objective which only predicts this entity, without the process of looking up its type and randomly choose the negative entity.
- I’m curious because I’m not sure how much effect do negative entities have. Although they are the same type of entities, it’s hard that the chosen entity is a strong negative entity. For instance, Barack Obama (Q76) is an instance of ‘human’, and I don’t think another entity of human that is randomly chosen is helpful. A similar observation was found in WikiHop dataset, which is a multi-choice QA dataset that all negative candidates are guaranteed to be the same type, but still question-candidates-only baseline (without context paragraphs) outperforms state-of-the-art models (https://openreview.net/forum?id=B1lf43A5Y7). It indicates that entities with the same type are not strong negative candidates.
- For this reason, I think it is possible that negative entities do not help, but the fact that the training objective is more entity-centric helps the performance on various different downstream tasks. That’s why I think the ablation with entry-centric masked LM can be good to see.
- Of course, this is just one of my hypotheses, and I believe the results in this paper are significant regardless, but I just wonder what authors think about this.

**Experience Assessment:**

I have read many papers in this area.

**Review Assessment: Checking Correctness Of Derivations And Theory:**

N/A

**Review Assessment: Checking Correctness Of Experiments:**

I assessed the sensibility of the experiments.

**Review Assessment: Thoroughness In Paper Reading:**

I read the paper thoroughly.

---

> ### Author Response · Authors · 2019-11-10
> **Reply to Review #2**
>
> Thank you for the positive feedback! Please see our explanations to your questions below:
>
> On terms of ablation models:
> BERT + 1M MLM indicates that we continue fine-tuning the base BERT model with only masked LM for additional 1M updates; and WKLM without MLM indicates fine-tuning BERT with only the knowledge learning, which is equivalent to “BERT+1M knowledge learning” since our WKLM method also relies on  BERT initialization.
>
> For WebQuestions, we split first construct the training samples (question, paragraph and answer span) with distant supervision. We split all the distant-supervised training data into train and validation splits (9:1). For TriviaQA experiments, we consider the open-domain setting. We directly use the code released by Lin et al. to generate the data and the statistics we reported are based on their code’s output. In Table 4, all the baselines and our method indeed consider the same open-domain setting, where a retrieval module is required to collect the paragraphs. Lee et al. (ORQA) also consider this setting despite the fact that they used a different retrieval model.
>
> Q1: It is true that some of the hyperlinks might not be accurate, but those cases are relatively rare and it is our goal to utilize this kind of weak supervision via large-scale pretraining.
>
> Q2: There could be several mentions that refer to the same entity. Our replacement strategy treats them individually: some of them are replaced while the others are not; the sampled negative entities could also be different.
>
> Q3: As clarified above, we consider entity mentions for replacement instead of entities. Thus, if an entity appears in multiple locations, it could be replaced multiple times.
>
> Q4: Yes, we use the relation “instance_of”. If an entity has multiple true types, we first random sample the type and then sample from the entities of that type.
>
> Q5: As some type could include a large number of entities, it is possible that some of the sampled entities are not hard negatives. However, compared to the language modeling objectives, which considers all the other false tokens evenly, our replacement strategy is still more likely to introduce strong negative signals, as we only distribute the probability to those entities with the same type. We have tried to introduce harder candidates by sampling from the entities within the same document and also with the same type. However, that strategy is not guaranteed to produce negative entities all the time and we did not observe any improvement.

---

> > ### Comment · AnonReviewer2 · 2019-11-10
> > **Thanks for your response**
> >
> > Thanks for your responses and clarification.
> >
> > Regarding ablation terms: That makes sense.
> >
> > Regarding WebQuestions setting: That resolves my concern. Hope the updated version of the paper describes the WebQuestions split statistics precisely in Table 2.
> >
> > Regarding TriviaQA setting: I am not sure if all models in Table 4 are in the same setting. For example, DSQA (Lin et al 2018) uses TriviaQA-unfiltered setting where Wikipedia documents & web search results from an external search engine are provided (which is part of released dataset). ORQA (Lee et al 2019) is an open setting where it discards all given documents and only assumes Wikipedia dump. Not sure about some of other baselines, e.g., original BERTSerini paper didn't evaluate on TriviaQA, I believe? Anyhow, it should be clear which setting was used in the paper.
> >
> > Regarding Q1: Thanks for the clarification, although I am not sure if they are rare to be ignored.
> >
> > Regarding Q2, Q3: That's exactly what I was curious, thanks.
> >
> > Regarding Q4: Got it, it would be nice if it is mentioned in the paper.
> >
> > Regarding Q5: I believe that many of the negatives would not be hard negatives. But I agree that it will be hard to generate hard negative examples. I think it would be even better if the paper admits it somewhere and describes some attempts to generate hard negative examples although it ultimately didn't work.
> >
> > Nonetheless, I think this paper is overall a strong paper and it looks like all reviewers vote for acceptance.

---

> > > ### Author Response · Authors · 2019-11-11
> > > **Further clarification on TriviaQA settings**
> > >
> > >
> > > Thanks for the detailed comments. We have revised the paper to add those detailed settings.
> > > Below is our further clarifications about the  QA experiments:
> > >
> > > The augmented version of BERTSerini (https://arxiv.org/pdf/1904.06652.pdf) also evaluates on TriviaQA and compares with DSQA, R^{3} and Evidence Aggregation. All the baselines are open-domain QA methods but the retrieval system and corpus might be different. For instance, BERTSerini use BM25 and Wikipedia, ORQA uses BERT-based retrieval and Wikipedia. The other methods and our method just make use of the retrieval results provided by the original TriviaQA paper (Bing Web search API and Web Documents). We will also make this clear in the revision.

---

### Official Review · AnonReviewer3 · 2019-10-22
**Official Blind Review #3**

**Rating:** 6

**Review:**

This paper proposes to trained better entity centric text embeddings by switching entities mentioned in the text to some other entities with the same type. The target is modeled as a binary classification task, which is trained jointly with the MLM loss. The authors do experiments on multiple tasks, and the model shows strong performance on all tasks. And the ablation study justifies that "knowledge pre-training" is crucial. The idea is novel and the experiment results suggest that the additional "adversarial" target helps. The writing is clear in general, but misses some implementation details.

A few questions:
1. In section 3.1, how do you rank the candidate answers with your model? Do you compute the logits in the same way as the baseline models?
2. In section 3.2.1, why do you use a different split for TriviaQA? Do you rerun the baseline models on this new split?
3. In section 3.2.1, the author claims that most answers in TriviaQA, SearchQA and Quasar-T datasets are entities. A interesting metric to evaluate would be how much improvements WKLM obtain on those questions, versus those whose answers are text spans.
3. In section 3.2.2, the authors mention that they use the CLS token to predict the entity types. How do you train the embedding your CLS token?
4. For the entity typing task in section 3.2.2, do you fine tune your model or it's evaluated in a zero-shot setting?



**Experience Assessment:**

I have published one or two papers in this area.

**Review Assessment: Checking Correctness Of Derivations And Theory:**

N/A

**Review Assessment: Checking Correctness Of Experiments:**

I assessed the sensibility of the experiments.

**Review Assessment: Thoroughness In Paper Reading:**

I read the paper at least twice and used my best judgement in assessing the paper.

---

> ### Author Response · Authors · 2019-11-10
> **Reply to Review #3**
>
> Thank you for your positive comments. Below are our answers to your questions:
>
> On candidate ranking: During WKLM pretraining, our model learns to predict a probability ($P(e|C)$) for each entity mention $e$ about whether the mention is correct (not replaced) or not (replaced). We directly use these predicted probabilities of entity candidates for ranking. We have added a description in the revision.
>
> On TriviaQA experiments: Our TriviaQA splits are consistent with the splits released by DSQA (Lin et al. 2018).  They reused the data processed by R3 (Wang et al., 2018a) and Evidence Aggregation (Wang et al., 2018b). Since the other systems are not open-sourced and the retrieval methods are different, we are not able to rerun those baselines.
>
> On Entity QA performance: Compared to SQuAD, these open-domain QA datasets are more entity-centric. We already described the ratios of entity answers in the paper. For SQuAD, our manual inspection on 100 random examples suggests that only around 40% of the answers are actually common entities. By looking at our model’s improvements over SQuAD and these open-domain QA datasets, we can see that our model is more effective at entity-centric questions.
>
> On CLS token: We prepend the [CLS] token to all sentences in the entity typing dataset and the embedding is updated during fine-tuning.
>
> On entity typing experiments: We fine-tune the model using the same training data as used by the compared approaches.

---

### Official Review · AnonReviewer1 · 2019-10-24
**Official Blind Review #1**

**Rating:** 6

**Review:**


This paper proposed to improve pre-training of language models (e.g. BERT) by incorporating information around entities based on English Wikipedia. The idea is very simple and straightforward: it takes all the anchor links from Wikipedia and replaces some entities by randomly sampling negative ones of the same entity type (according to Wikidata) and adds an extra binary prediction task which predicts if the entity has been replaced or not.

The model was initialized by BERT (or the authors’ BERT reimplementation) and trained for another 1M steps with the new training objective and reduced % of masking tokens.

The model was evaluated on a fact completion task (created by the authors on the 10 sampled Wikidata relations) and several open-domain QA datasets and an entity typing dataset FIGER, and achieved significant improvements on the BERT baselines.

Overall, I think this is a strong paper. The idea is simple but effective, the experiments are thorough and improvements over the BERT baselines are significant.

Below are some concerns I had when I read the paper and also some suggestions on how to improve this paper:

1) I am slightly concerned about the evaluation of the fact completion task and its baselines.

- Why are there only 190-906 candidates for these relations? How were the candidates chosen? Why not use the full set of possible candidates of that entity type?

- I am not sure why you picked the most common entities for predictions. Fact completion for rare entities would be more challenging and practical. Also, the models might favor choosing more common entities as well.

- I am also not sure if the BERT baseline (by using k [MASK] tokens when the candidate answer has k tokens and taking the average of the k probabilities) is a strong one or not in this setting, as BERT was not trained in this way and it is unclear if this would make BERT favor shorter entities or not.

2) OpenQA results (Table 4): there is a very strong baseline coming out recently (an EMNLP’19 paper):

Multi-passage BERT: A Globally Normalized BERT Model for Open-domain Question Answering.

Even for their BERT-base model, TriviaQA F1 was 67.5, SearchQA F1 was 70.6 and Quasar-T F1 was 59.0. It is okay to not directly compare to their results (the focus is different), but the authors should be aware of their results and perhaps remove the state-of-the-art claim.

3) I’d be interested in seeing more ablation studies on the importance of masking/replacement choices. What is the percentage of entities that have been replaced? 50%? The only thing I can find is that no adjacent entities have been replaced at the same time. How important is that?  I imagine that the percentage of entities that have been replaced should also matter the performance significantly.

4) If I understand correctly, the model was first trained (as BERT) on English Wikipedia + BooksCorpus and then later trained only on English Wikipedia. I wonder how important the first stage would still be. Could add an experiment that trains on Wikipedia only?

Minor suggestions:
1) Please use “English Wikipedia” instead of “Wikipedia” (#BenderRule)
2) Table 1: don’t put “572” next to “Average Hits @10”. It is confusing.


**Experience Assessment:**

I have published in this field for several years.

**Review Assessment: Checking Correctness Of Derivations And Theory:**

I carefully checked the derivations and theory.

**Review Assessment: Checking Correctness Of Experiments:**

I carefully checked the experiments.

**Review Assessment: Thoroughness In Paper Reading:**

I read the paper thoroughly.

---

> ### Author Response · Authors · 2019-11-10
> **Reply to Review #1**
>
> Thank you for the detailed comments, we have updated the paper accordingly. Please see our reply below:
>
> On Fact Completion Evaluation:
>
> Candidate selection: For each relation, we use the groundtruth answer entities from all queries as the candidate set. Using the full set of entities of the same type could result in much larger candidate sets (there are more than 5 million person entities), which makes it computational expensive to evaluate each query. Note that we view the fact completion task as a probing task instead of a real-world application. Thus, we choose to use the compact candidate sets for fast evaluation.
>
> Why selecting common entities for evaluation: We focus on common entities since common world knowledge is likely to be more useful to NLP applications. Automatically completing knowledge bases with rare long-tail entities is an important problem, but that’s not the primary focus of this work.
>
> BERT baselines: BERT is not trained to handle multi-token entities, so using the average token probability is the best BERT baseline we could think of for the fact completion task. Our further inspection shows that the BERT baseline does not favor single-token entities: the average ratio of single-token candidates is 24.5% while the ratio of single-token entities among the BERT’s top10 predictions is 20.0%.
>
> On OpenQA results:
> Thank you for pointing us to the new model (Wang et al. EMNLP’19, which is presented after the submission deadline). Wang et al. show the benefits of normalizing the prediction probabilities across multiple passages during training. As stated by the reviewer, our focus is different: instead of investigating more effective ways to use pretrained models for QA, we focus on improving the pretrained model itself and our model can easily stack with the new method. Thus, we use a quite straightforward method for the QA experiments. We have included a discussion in the revision.
>
> On entity replacement strategy:
> With the current replacement strategy, the replacement ratio is approximately 50%. With limited resources, we have tried replacing all entities and also increasing the margin to 2 between replaced entities. These two model variants are able to match vanilla BERT’s performance on SQuAD (~89.0F1 and ~90.0F1 compared to 90.5F1 with BERT) but fail to produce as much improvement as our final model. These numbers suggest that the new training objective will be less effective if the replacements are too sparse or too dense. Also, as we are doing joint training (with MLM), the performance of the two variants is still able to match the vanilla BERT.
>
> On two-stage training:
> We designed our two-stage method in view of the practices of several existing work (https://arxiv.org/abs/1905.07129, https://arxiv.org/abs/1909.04164,https://arxiv.org/abs/1905.03197). The BERT initialization allows us to start with a strong baseline and the second stage training is also very stable after the initialization. As the pretraining + end tasks takes a long time to finish, we are unable to add the ablation during the discussion period due to resource constraints. We are happy to add those experiments in the next revision.

---

### Decision · Program_Chairs · 2019-12-19

**Decision:**

Accept (Poster)

**Comment:**

This submission proposes a secondary objective when learning language models like BERT that improves the ability of such models to learn entity-centric information. This additional objective involves predicting whether an entity has been replaced. Replacement entities are mined using wikidata.

Strengths:
-The proposed method is simple and shows significant performance improvements for various tasks including fact completion and question answering.

Weaknesses:
-The experimental settings and data splits were not always clear. This was sufficiently addressed in a revised version.
-The paper could have probed performance on tasks involving less common entities.

The reviewer consensus was to accept this submission.